



# Full scale deformation measurements of a wind turbine rotor in comparison with aeroelastic simulations

Stephanie Lehnhoff[1], Alejandro Gómez González[2], and Jörg R. Seume[1]

[1]ForWind, Institute of Turbomachinery and Fluid Dynamics - Leibniz Universität Hannover
[2]Siemens Gamesa Renewable Energy A/S

**Correspondence:** Stephanie Lehnhoff (lehnhoff@tfd.uni-hannover.de)

**Abstract.** The measurement of deformation and vibration of wind turbine rotor blades becomes highly important as the length of rotor blades increases with the growth in demand for wind power. The requirement for field validation of the aeroelastic behaviour of wind turbines increases with the scale of the deformation, in particular for modern blades with very high flexibility and coupling between different vibration modes. However, performing full-scale field measurements for rotor blade deformation is not trivial and requires high temporal and spatial resolution. A promising deformation measurement technique is based on an optical method called Digital Image Correlation (DIC). A system for the application of DIC for full field measurements of wind turbine rotors has been developed and validated in the past years by ForWind, Institute of Turbomachinery and Fluid Dynamics, Leibniz Universität Hannover. The whole rotor of the wind turbine is monitored with a stereo camera system from the ground during measurement. Recently, DIC measurements on a Siemens Gamesa SWT-4.0-130 test turbine were performed on the tip of all blades with synchronized measurement of the inflow conditions by a ground-based LiDAR. As the turbine was additionally equipped with strain gauges in the blade root of all blades, the DIC results can be directly compared to the actual prevailing loads. In the end, the measured deformations are compared to aeroelastic simulations.

The deformation measured with DIC on the rotor blade tips shows the same qualitative behaviour when compared to loads measured with strain gauges in the blade root. This confirms that the DIC measurements correlate with the prevailing loads in reality. The comparison with aeroelastic simulations shows that the amplitude and trend of the in-plane deformation is in very good agreement with the DIC measurements. The out-of-plane deformation shows slight differences, which could be caused by the difference between real wind conditions and the wind statistics on which the simulations are based. The combined rotor blade pitch and torsion angle measured with DIC is in good agreement with the actual pitch value of the turbine. A detailed comparison with aeroelastic simulations shows that the amplitude of torsion measured with DIC is higher which might be caused by an inaccuracy of the experimental setup. This will be focus of future work. All in all, DIC shows very good agreement with comparative measurements and simulations which shows that it is a suitable method for measurements of deformation and torsion of multi-megawatt wind turbine rotor blades.

**Keywords.** full scale measurements, wind turbine rotor blade deformation, digital image correlation



## 1 Introduction

The increasing demand in reduction of costs per kWh of wind turbines results in ever larger rotor diameters. In recent years, this has led to a crucial change in the structural design of wind turbine blades, as the relative mass of the blade needs to decrease for a product to be economically viable. The knowledge of the aeroelastic behaviour is one of the biggest challenges in today´s wind turbine engineering and especially in the future (Veers et al. (2019)).

Along with this comes the need for experimental validation of aeroelastic modelling. Until today, the load of wind turbine
rotor blades is usually measured with strain gauges in edgewise and flapwise directions, however measuring the rotor blade deformation and torsion is still a challenge. Optical measurement methods can make a contribution to this. In the past, several optical measurement methods were successfully applied on full-scale wind turbines for the determination of rotor blade deflections during operation (Schmidt Paulsen et al. (2009), Ozbek and Rixen (2013), Grosse-Schwiep et al. (2014), Lutzmann et al. (2016)).

For the direct measurement of rotor blade deformation (in-plane as well as out-of-plane) and torsion of full-scale wind turbines, only a few suitable measurement methods exist which are shortly introduced. In the past years, Siemens Gamesa developed an in-house photographic method for the detection of these variables (Mayda et al. (2013)). For this method, a camera is installed in the blade root region of a rotor blade, facing the blade tip. At different radial positions, optical markers are installed upright relative to the pressure side and on these sections, the deformation and twist can be detected. Another
method called BladeVision was developed by SSB Wind Systems (Nidec SSB Wind Systems GmbH (2020)). For this method, a camera is installed in the inner side of the blade root region of the blade. Reflectors are also installed inside at different radial positions and are monitored by the camera for the determination of deformation and twist on those positions. Another method was developed by ForWind, Institute of Turbomachinery and Fluid Dynamics, Leibniz Universität Hannover. This method is based on Digital Image Correlation (DIC).

For this method, a stereo camera system is installed in the area in front of the turbine and certain speckle patterns are applied on the blades' pressure side. On those sections where the pattern has been applied, the deformation and torsion of the rotor blade can be detected. Before DIC was applied at full scale wind turbines, the feasibility and accuracy was determined on a scaled wind turbine model (Winstroth and Seume (2014a)) as well as within a fully virtual experiment (Winstroth and Seume (2014b)). Afterwards, the feasibility of this measurement method at full scale was proven (Winstroth and Seume (2015)) and
a comparison of out-of-plane deformations (measured with DIC) with aeroelastic simulations was also done (Winstroth et al. (2014)). The results show that such a comparison is not trivial, as the selection of time series can have a significant influence on the results. Regardless, these measurements are still a valid tool for the validation of aeroelastic codes. All these three methods have advantages and disadvantages. As an example, DIC is sensitive to the changing weather conditions outside, but it can be easily installed on the blade compared to the other two methods.

This paper shows results of DIC measurements on a wind turbine in the field and a comparison with aeroelastic simulations. Firstly, the experimental setup for the execution of optical measurements on a full scale wind turbine is described. The speckles for DIC were applied in the tip region on all three blades. For the detection of the movement and rotation of the hub, three big



speckles were applied on the hub itself. The hub movement is later on used for the determination of in-plane deformation (IP), out-of-plane deformation (OoP) and torsion out of the DIC signal. Afterwards, the functionality of the two optical methods applied, DIC and Marker Tracking, are briefly explained. Measurement results are shown and compared to strain gauge signals at the blade root for a qualitative experimental validation of the optical method. For a rough validation of the combined rotor blade pitch and torsion angle, the DIC signal will be compared to the pitch signal of the turbine. This can only prove a trend, as there is no additional measurement technique installed for the determination of rotor blade torsion. Results of a Fast Fourier Transformation (FFT) of the signals will be shown and compared to the vibrational eigenfrequencies that are expected to occur from numerical investigations. Finally, the DIC measurements will be compared to aeroelastic simulations of the turbine as a first try for an experimental validation of rotor blade deformation and torsion, based on DIC measurements.

## 2 Experimental Setup

The general measurement setup for the execution of DIC measurements on wind turbines is shown in Figure 1. Different radial positions of the blades can be equipped with self-adhesive foils to build a speckle pattern on the rotor blade. The deformation can be captured in those areas where the speckles are applied, so this could be done along the whole length of the blade. A stereo camera system is placed in the area upstream of the turbine to monitor the whole rotor during operation.

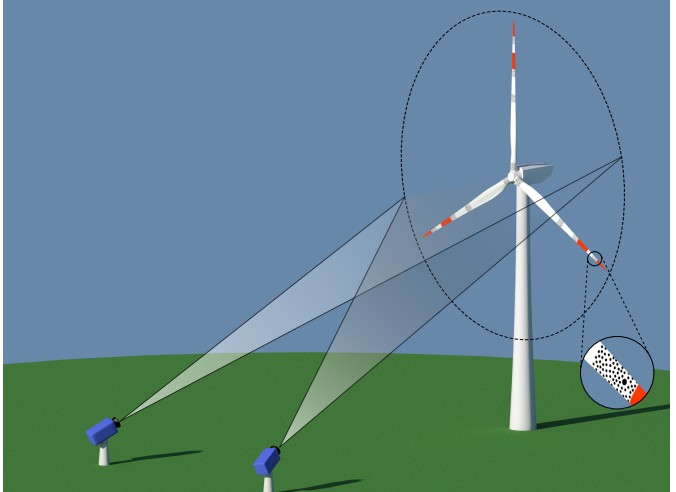

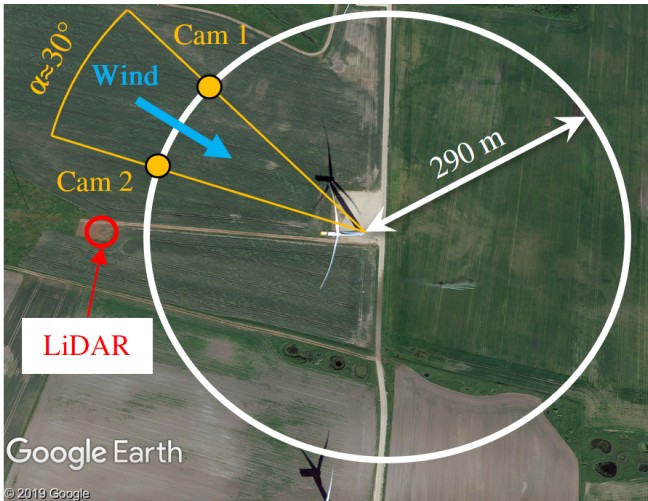

**Figure 1.** Schematic of a camera setup in front of a full-scale wind turbine. The magnified cutout near the blade tip shows the random black-and-white pattern on the pressure side (Winstroth and Seume (2015)).

**Figure 2.** Experimental setup at the Høvsøre wind turbine test site.

In this measurement campaign, a Siemens Gamesa SWT-4.0-130 test turbine located at the DTU wind turbine test center in Høvsøre in Denmark, was equipped with a speckle pattern for DIC measurements in the tip region of all three blades. The





cameras have a resolution of 25 MPx, and take pictures simultaneously with a frame rate of 30 fps. Each camera is connected
via CameraLink to a measurement computer to store the pictures directly on the hard drive. Due to the high data rate of
750 MBps per camera, the maximum measurement duration with the current setup is limited to 10 minutes. Usually it is not
the hard drive which limits the measurement duration, rather the change in weather and ambient lighting conditions.

Each camera is equipped with a lens of a 58 mm fixed focal length. In order to monitor the full rotor diameter of 130 m,
the cameras are placed 290 m away from the foundation of the turbine, see Figure 2. The cameras are positioned in a 30 °
stereo angle configuration relative to the wind turbine. The wind direction should remain constant in the region between the
two cameras. The wind speed and direction at 10 different heights throughout the full extension of the rotor is measured at a
sampling frequency of 1 Hz with a LiDAR located at a distance of 2.5 rotor diameters in front of the turbine in order to be able
to assess shear and veer properly. Furthermore, the atmospheric temperature, pressure, and humidity are also logged.

The speckles were applied on the pressure side of the blades from a lift in the range from 55 m to 60 m measured from the
blade root, see Figure 3. The speckle pattern needs to be different for all blades, as DIC finds a unique greyscale signature for
every measurement point. Per blade, approximately 50 speckles with a diameter of 20 cm were applied that build a random
black-and-white speckle pattern. The hub is used to define the rotor plane and the rotational axis. In this case, three single dots
with a diameter of 70 cm were applied, which are analyzed with a Marker Tracking algorithm (see Figure 4).

The turbine is instrumented with strain gauges in the root of all three blades and furthermore, the following operational
parameters are logged: pitch, rotor speed, and power with a 25 Hz sampling frequency. This is useful for a comparison of DIC
with conventional measurement methods in the field.

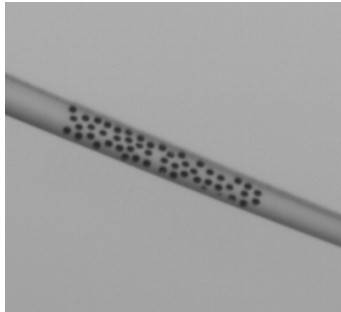

**Figure 3.** Random speckle pattern for the application of DIC on the blade tips.

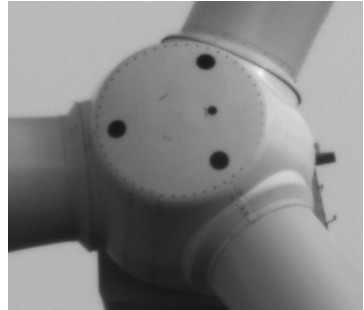

**Figure 4.** Three big dots on the nacelle for the application of Marker Tracking on the hub.

DIC is typically used in laboratory environments with constant illumination. As this setup is now applied in the field where
the sun is the only suitable light source, a great deal of experience is required to perform a successful measurement under these
conditions. The movement of clouds makes it a challenge to find a time slot which is longer than 5 minutes where illumination
conditions remain constant.

The difference between DIC and Marker Tracking is that with Marker Tracking the single dots are tracked and not the full-
field area between them. The advantage is that the position of the single markers is clearly defined, which comes along with the



disadvantage of a higher inaccuracy. Out of this setup, the track of three single dots is extracted to define the rotor axis and the rotor plane. The blade tip regions are evaluated with the DIC algorithm and result in an areal information of the whole speckle
100 pattern area, defined by the track of approximately 1000 points per blade tip.

## 3 Digital Image Correlation

In general, the term Digital Image Correlation describes an optical measurement method which is part of photogrammetry, that acquires images to calculate the full-field shape, deformation and/or motion measurements of certain objects (Sutton et al. (2009)). This process consists of the digital image acquisition itself, the storage and the performance of an image analysis to
105 obtain motion and deformation out of the images. In this part, the analysis will be briefly described. The reader is referred to Sutton et al. (2009) for a more detailed description of the analysis methods.

 The DIC algorithm applies several different analysis methods. It all starts with the recognition of the same points in all images. This process is shown in Figure 5 for one measurement point.

 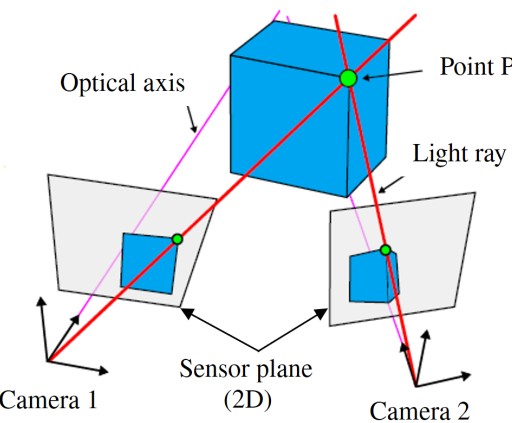

**Figure 5.** DIC algorithm finds points by tracking the greyscale signature of the subset in every picture.

**Figure 6.** The position of the point P in 3D is calculated by the position of the two cameras relative to the wind turbine, and the individual position of the measurement point in the left and right picture.

 The white dot in the middle of the green rectangle is the actual measurement point, which is defined by the greyscale
110 signature $F$ of the neighbouring pixels, that form a subset (green rectangle). At the time $T1$, the measurement point in image 1 of the left camera (usually the reference picture), is defined and is found in the following pictures through the use of a





correlation function of the greyscale value of the subset. This correlation function is based on the normalized cross-correlation criterion, that is actually the origin of the term "correlation" in DIC.

$$\chi^2_{NCC} = \frac{\sum FG}{\sqrt{\sum F^2 \sum G^2}} \tag{1}$$

The correlation criterion $\chi$ is bounded in the interval [0, 1], where 1 represents the perfect match. $F$ refers to the greyscale value of the subset in the reference image 1, and $G$ refers to the greyscale value in the following pictures, e.g. image 2. Usually, the maximum match is found under application of the Levenberg-Marquardt algorithm (Levenberg (1944), Marquardt (1963)).

This criterion is extended to account for lighting offsets and scales relative to the reference picture, and results in the zero-mean normalized sum of squared difference criterion (ZNSSD):

$$\chi^2_{ZNSSD} = \sum \left( \left( \frac{\sum \bar{F}_i \bar{G}_i}{\sum \bar{G}_i^2} G_i - \bar{G} \frac{\sum \bar{F}_i \bar{G}_i}{\sum \bar{G}_i^2} \right) - F_i + \bar{F} \right)^2 \tag{2}$$

The reference subset is defined in the reference image 1 of the left camera and is usually rectangular. To find a similar (or in perfect case the same) greyscale in the following pictures of the same camera, a shape function needs to be introduced, as the subset might not have the same shape. This shape function $\xi(x,p)$ is used to transform pixel coordinates in the reference subset into coordinates in the image after deformation. $x$ represents the position of the pixel in the image and $p$ represents the

displacement. This results in a correlation function that is dependent on the shape of the subset as follows:

$$\chi^2(p) = \sum (G(\xi(x,p)) - F(x))^2 \tag{3}$$

For an affine transformation, $\xi$ can be defined as:

$$\xi(x,p) = \begin{bmatrix} p_0 \\ p_1 \end{bmatrix} + \begin{bmatrix} 1 + p_2 & p_3 \\ p_4 & 1 + p_5 \end{bmatrix} x \tag{4}$$

The stereo matching between the left and the right camera is done under application of a plane-to-plane homography matrix.

This homography matrix relates image coordinates to coordinates on a plane in space. As the coordinates of both cameras in the world is known after the external calibration, image coordinates of the left camera can be related to image coordinates of the right camera through a homographic transformation, more commonly referred to as rectification in computer vision.

To achieve the maximum geometrical resolution, sub-pixel interpolation is applied in the matching algorithm. The sub-pixels are interpolated by a continuous 8-tap spline.

The result of the application of the DIC algorithm with the software Vic3D by Correlated Solutions, Inc. (Correlated Solutions, Inc. (2020) can be seen in Figure 7. The measurement points that are obtained are highlighted in green. It can be seen that the algorithm did not converge in the outer region, which is due to the definition of the subset size.





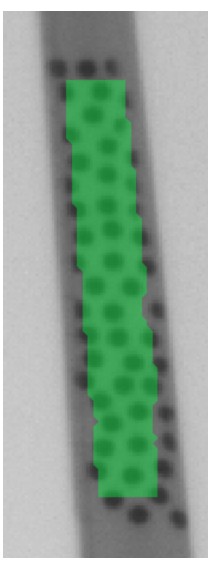

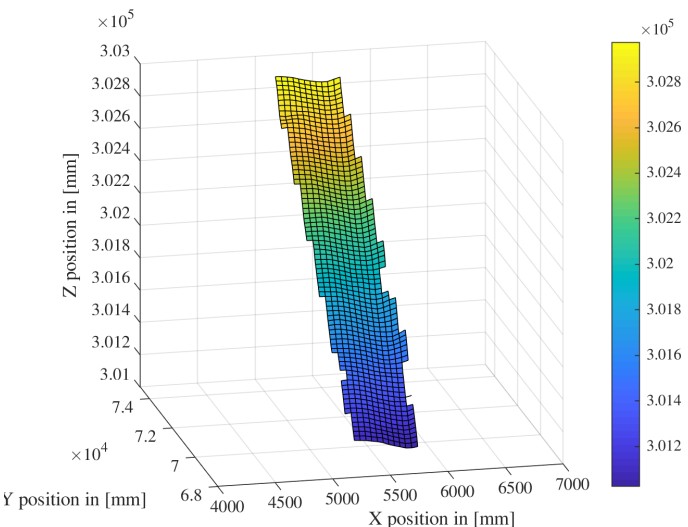

**Figure 7.** Measurement points plotted on the rotor blade

**Figure 8.** Measurement points plotted in 3D, not aligned to the rotor coordinate system

This data set can be directly imported in Matlab, the result of which is shown in Figure 8. The measurement points are not aligned to the rotor coordinate system at this evaluation step.

The Marker Tracking algorithm also applies a sub-pixel interpolation to find the right match of the marker in all images, however there is no definition of subsets, as the marker itself is directly tracked. This guarantees that the exact position of the markers will be calculated, but results in a less accurate signal compared to the areal DIC method. This requires the application of a low pass filter to the hub track. As the movement of the nacelle is slow compared to the movement of the blade tips, this method is still suitable for the detection of the hub track. The areal DIC tracks are not filtered and thus can be used directly for 145 evaluation.

## 4 Determination of rotor blade deformation and torsion

The evaluation of the optical measurement is split into five main parts, which is shown in Figure 9. The first part **(1)** is the detection of the positions of the speckle pattern out of the pictures, i.e. the application of a DIC algorithm to the image series. This is done with the commercial software Vic3D from Correlated Solutions, Inc (Correlated Solutions, Inc. (2020)). The 150 software can track the position of the speckle regions even under a rotating movement of the object. In a second step **(2)**, the movement of the hub is determined by tracking the position of the three markers on the hub itself with a Marker Tracking algorithm, which is also included in Vic3D. This defines the rotor axis as well as the rotor plane **(3)**, which is necessary for the next step. In a fourth step **(4)**, the positions are classified into in-plane deformation (IP) and out-of-plane deformation (OoP) by





removing the hub movement from DIC data. The OoP deformation can be further used to calculate the torsional deformation

of the rotor blade (**5**).

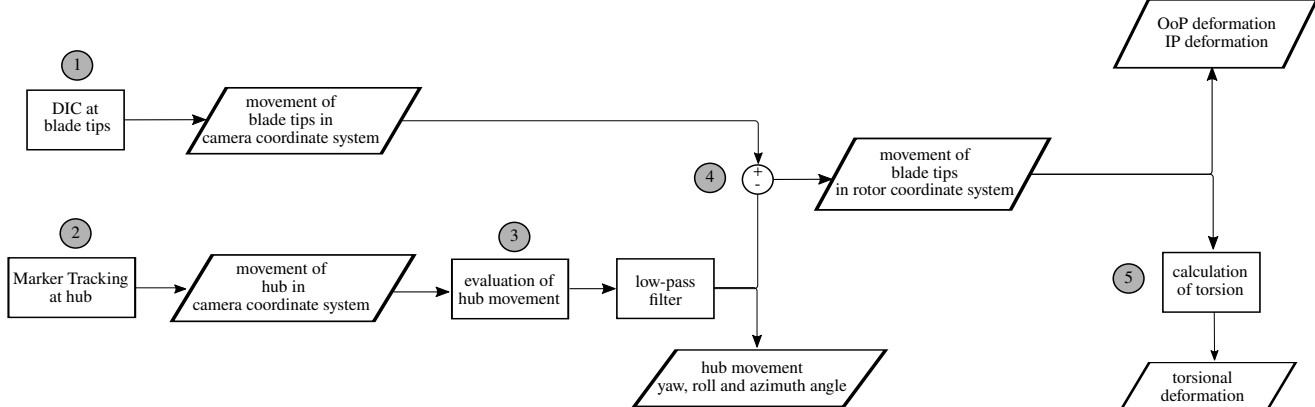

**Figure 9.** Evaluation of DIC and Point Tracking method for determination of rotor blade deformation and torsion

The output of DIC is a full-field information of the position of the surface of the rotor blades' pressure side in 3D. An example for the direct output of one DIC measurement point is shown in Figure 10. The coordinate system is not aligned, thus the measurement point rotates around an undefined rotational axis. A change in the yaw position of the rotor can be clearly seen in this track. To define the deformation in OoP and IP, the rotor plane needs to be aligned with the rotor coordinate system.

To define the position and orientation of the rotational axis, which is step (**3**), the tracks of the markers on the hub are used. This is done in three steps:

i) Translational alignment of the hub center to the origin of the coordinate system

→ elimination of translational movement

ii) Rotational alignment of the normal vector on the hub to the x-axis of the coordinate system

→ elimination of yaw and roll angle

iii) Rotational alignment of the measurement point around the rotational axis

→ elimination of azimuth angle

The translational displacement (**i**) of the rotational axis is found by determining the center of the position of the three markers. The markers are not perfectly positioned at the same distance to the rotational axis, which results in a remaining

rotational radius of approximately 20 mm, which can safely be considered negligible. The translational displacement of the rotational axis is determined for every time step and removed from the original DIC data.

The rotational misalignment (**ii**) between the normal vector of the rotor plane and the x-axis in the coordinate system is determined. This results in two angles which are removed from the DIC data for every time step: yaw and roll angle. The result of this can be seen in Figure 11. What remains is the rotation around the x-axis, which is defined as the azimuth angle.



In a third step, the azimuth angle needs to be removed **(iii)**. For this, the reference measurement point of the center of rotation is aligned with the z-axis. The azimuth angle is determined as the rotational offset around the x-axis between the actual measurement point and the reference point. Now that the azimuth-angle is determined, it can be removed from the DIC signal that is aligned with the rotor plane.

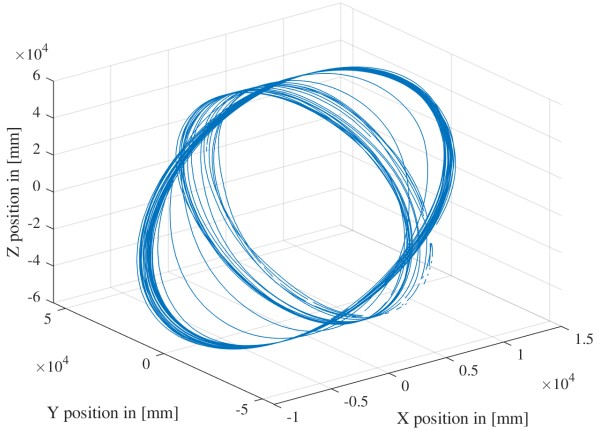

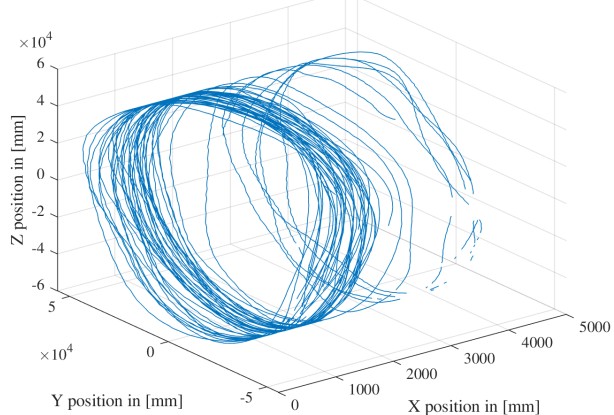

**Figure 10.** Track of one measurement point of DIC withouth alignment

**Figure 11.** Track of one measurement point of DIC aligned to the rotor plane

    The displacement that remains in the DIC measurement points is defined as follows:

– movement in x-direction → out-of-plane deformation (OoP)

    – movement in y-direction → in-plane deformation (IP)

    – movement in z-direction → radial deformation

    The radial deformation is affected by the radial displacement of the hub marker center to the actual rotational axis remaining in the measurement data.

Figure 12 shows a view of the rotor blade chord of length $c$ in relative position to the rotor plane. The angle $\beta$ defines the rotation around the vertical axis of the rotor blade, and is a combination of pitch and torsion angle. This angle can be determined by the position of points between the leading edge $P_{LE}$ and the trailing edge $P_{TE}$.

$$\beta = \arcsin\left(\frac{x(P_{TE}) - x(P_{LE})}{c}\right) = \arcsin\left(\frac{dx}{c}\right) \tag{5}$$

    Depending on the value of $c$, the resolution of the OoP position may need to be very accurate to determine the torsion angle.

If $c$ has a value of 700 mm and $\beta$ should be determined with a resolution of $0.1°$, then $dx$ needs to be resolved with an accuracy of 1.2 mm.





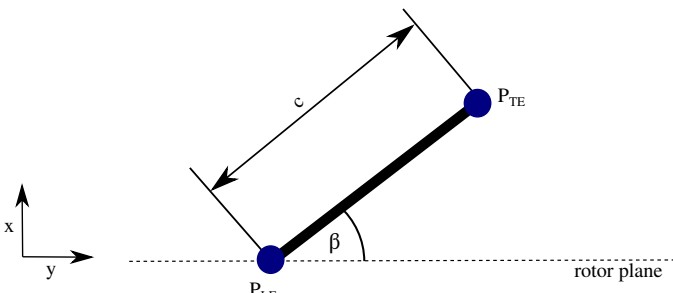

**Figure 12.** Determination of the pitch + torsion angle $\beta$ out of DIC measurements; c: chord length of the rotor blade, x: OoP position, y: IP position

## 5   Results

This section contains measurement results of one DIC measurement time series and related simulations. The DIC measurement duration was five minutes and the simulations were done for a ten minute time series, reproduced by the mean wind conditions during the time slot. For the simulations, the aeroelastic solver BHawC (Siemens Gamesa in-house aeroelastic solver (Rubak and Petersen (2005), Skjoldan (2011))) is used. Usually, a minimum of six and a maximum of 20 simulations with the same mean wind conditions, but under variation of turbulence seeds, are conducted for a time series. In this case, nine simulations were conducted to become independent of the influence of turbulent seeds.

The mean measured wind conditions during the time period are shown in Figure 13. These conditions were measured by the ground-based LiDAR. The azimuth angle in the following diagrams is defined according to Figure 14. All following results for measurements and simulations were extracted at a radial distance of approximately 56.5 m distance to the blade root.

| Property | Value |
|---|---|
| Air density | $1.226\ kg/m^3$ |
| Wind speed | $16.48\ m/s$ |
| Average yaw error | $-4.28\ °$ |
| Turbulence intensity | $0.0693\ \%$ |
| Average shear coefficient | $0.07865$ |
| Simulation duration | $600\ s$ |

**Figure 13.** Measured wind conditions as input for simulations

**Figure 14.** Definition of azimuth angle





## 5.1 Rotor blade deformation

Figures 15 to 18 show the output of processed DIC measurements on all blades. In the OoP time series the influence of a continuous change in pitch angle can be clearly seen. At the beginning of Figure 16 the influence of an asymmetric flapwise vibrational mode can be seen on all rotor blades for a few seconds before the vibration subsides. A direct comparison of OoP and IP deformation shows that the amplitude of IP deformation is higher compared to OoP.

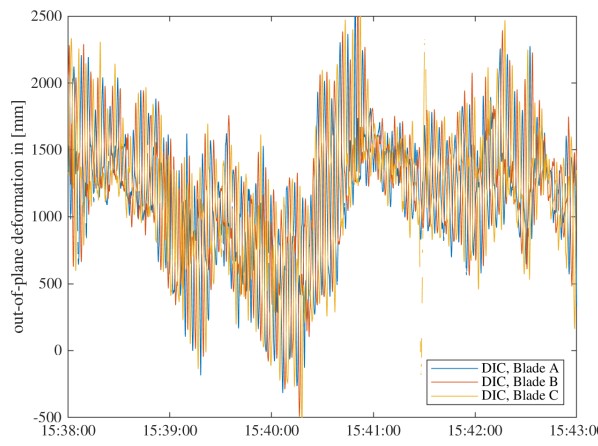

**Figure 15.** OoP DIC signal of all blades

**Figure 16.** OoP DIC signal of all blades - Zoom

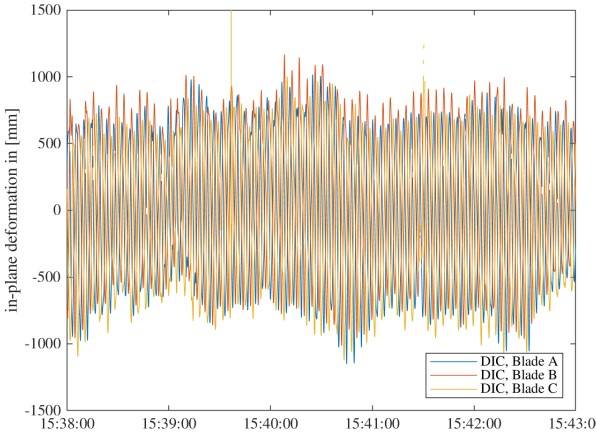

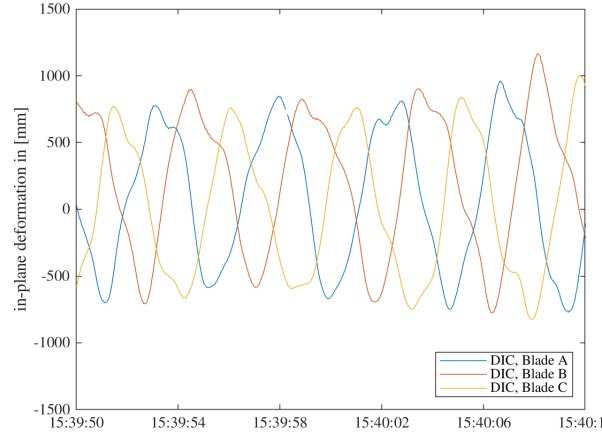

**Figure 17.** IP DIC signal of all blades

**Figure 18.** IP DIC signal of all blades - Zoom





Figure 19 shows a qualitative comparison of the flapwise bending moment in the blade root and the deformation at the blade tip in OoP direction.

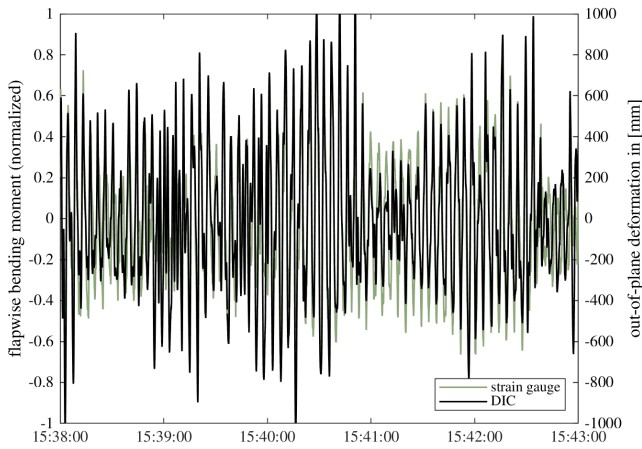

**Figure 19.** Qualitative comparison of measured flapwise bending moment and OoP deformation of Blade B

Both variables have been reduced by their moving average for an improved comparability. It can be seen that the qualitative
behaviour of both variables is nearly identical which confirms that the OoP deformation of rotor blades, measured with DIC, corresponds to the flapwise loads prevailing in reality. The same behaviour can be shown for IP deformation in comparison with the edgewise bending moment.

**Comparison with aeroelastic simulations**


In Figure 20 and 21 OoP and IP deformation measured with DIC is shown in direct comparison with simulation results. At first sight, the OoP DIC signal is in good agreement with the simulations, while the IP DIC signal shows a slightly lower amplitude compared to single simulations.

For a better comparison over the whole time series, the deformation is plotted against the azimuth angle. In Figures 22 and
23 the mean results for OoP deformation against azimuth angle are shown. The values are divided into 1° steps of azimuth angle and the values for the corresponding mean value and standard deviation are obtained.



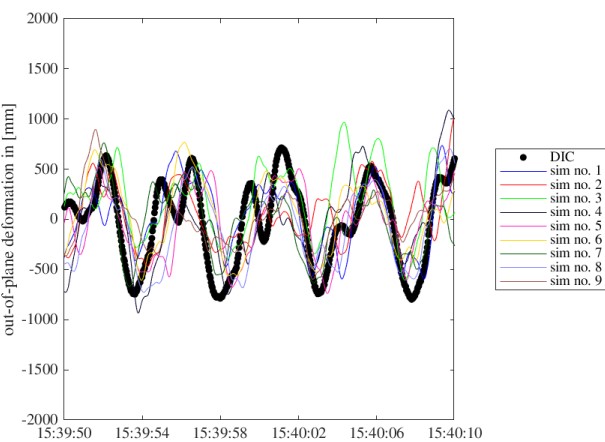

**Figure 20.** Comparison of OoP deformation measurement and simulations of Blade B- short time series

**Figure 21.** Comparison of IP deformation measurement and simulations of Blade B- short time series

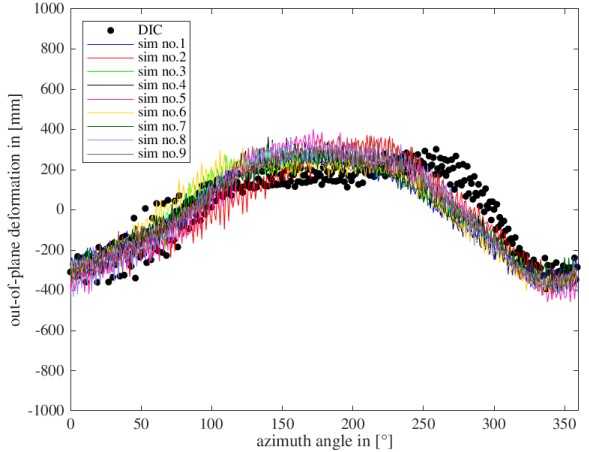

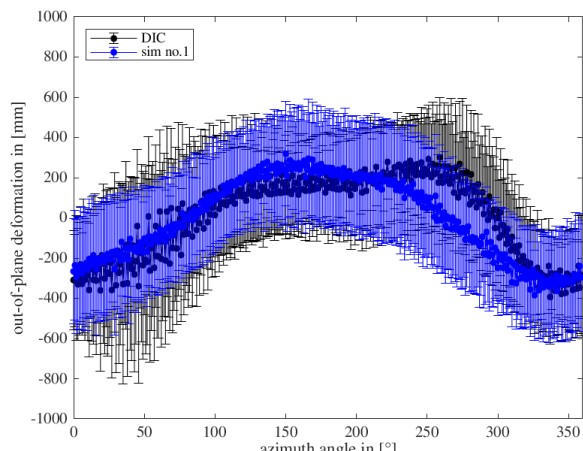

**Figure 22.** Comparison of OoP deformation of DIC and simulations of Blade B - mean values

**Figure 23.** Comparison of OoP deformation of DIC and simulation no.1 of Blade B - with standard deviation

In general, the OoP deformation measured with DIC is in very good agreement with the simulations. The maximum OoP deformation for simulations of Blade B appears at approximately 180 °, while for DIC it is shifted and appears at around 250°. One reason for this could be an influence of the actual prevailing weather conditions. The simulations are based on statistics, while the measurements are a result of real wind conditions which can be different in this case. The standard deviation for both simulations and DIC are in good agreement, as can be seen in Figure 23.






The IP deformation of DIC is in very good agreement with the simulations, which is shown in Figures 24 and 25. The location of minimum and maximum IP deformation is nearly the same and so is the amplitude and the related standard deviation.

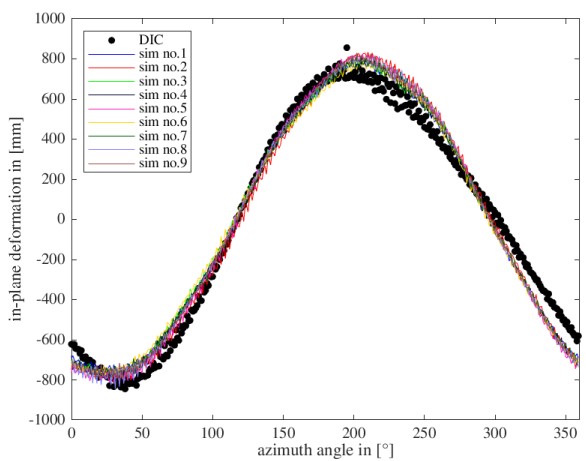
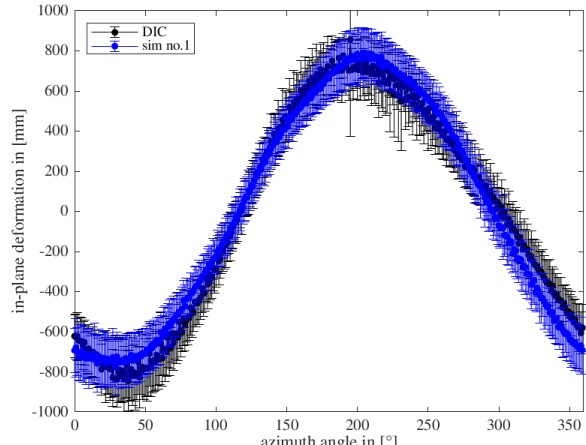

**Figure 24.** Comparison of IP deformation of DIC and simulation no. 1 of Blade B - whole time series

**Figure 25.** Comparison of IP deformation of DIC and simulations of Blade B - whole time series

## 5.2 Rotor blade torsion

A result for the combination of rotor blade pitch and torsion angle measured with DIC is shown in Figure 26. The DIC signal follows clearly the pitch signal of the turbine for all three blades. In comparison with results from simulation no. 1, as shown in Figure 27, the amplitude of measured torsion is higher. In reality, the pitch signal has a higher range (from 17 ° to 11°) compared to simulation no. 1 (from 14° to 11°), but this can not explain the difference between measured and simulated torsion.

A direct comparison between measured and simulated torsion is shown in Figure 28. The moving average has been removed
from all data sets and shows clearly, that the torsion measured with DIC is higher compared to simulations, but generally shows the same trend. This becomes even clearer when the torsion is plotted against deformation, as shown in Figure 29. The trend of the coupling between rotor blade torsion and OoP deformation can be reproduced from the DIC measurements, but with a significantly higher amplitude.

The reason for this difference could lie either in an inaccuracy of the simulation or of the DIC measurement. However, if
the torsion would have an amplitude as high as that measured with DIC, the actual measured loads would be expected to give significantly different values, too. As this is not the case, the dynamic amplitude of torsion measured with DIC is physically not plausible in this case. The reason for this is not clearly identified by now and will be further investigated. As it is known from past measurement campaigns, the spatial resolution and thus the accuracy of DIC can be improved by increasing the number of speckles on the blades, what will be taken into account for future measurement campaigns.





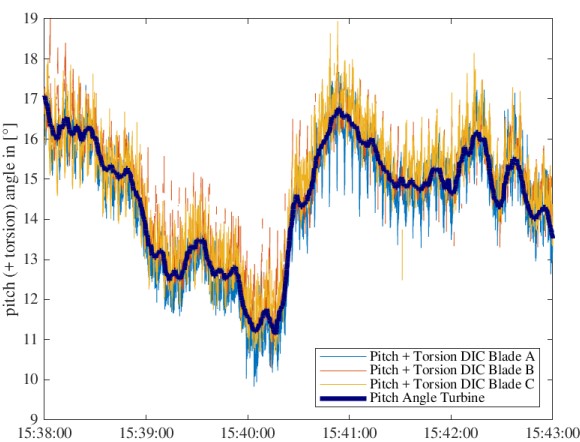

**Figure 26.** Masured wind turbine pitch angle and DIC Pitch + Torsion angle

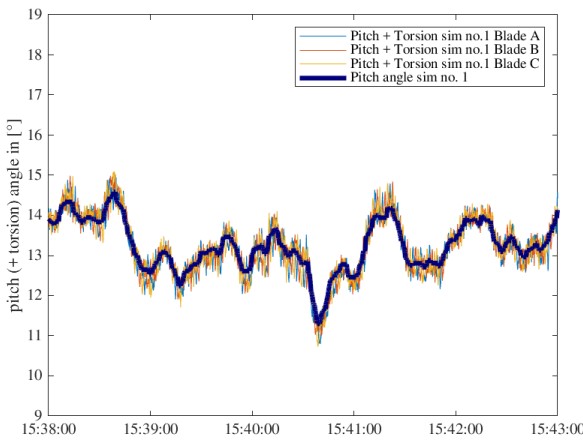

**Figure 27.** Simulated Pitch + Torsion angle of simulation no. 1 of all blades

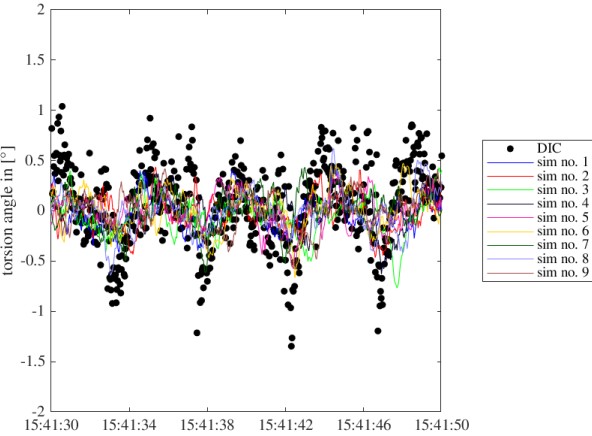

**Figure 28.** Comparison of torsion angle of Blade A of DIC and simulations - short time series

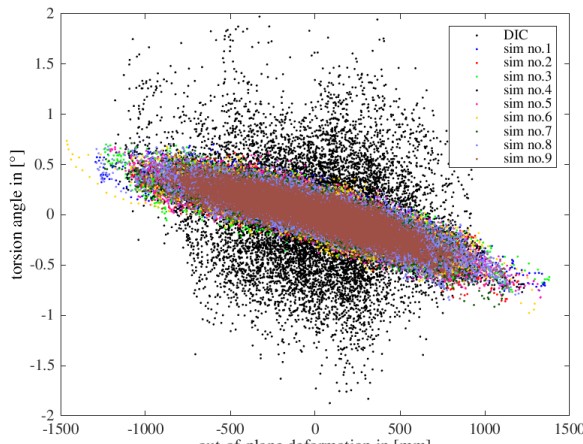

**Figure 29.** Coupling of rotor blade torsion and OoP deformation

## 5.3   FFT

Lastly, deformations measured with DIC are compared to simulations and bending moments in the frequency spectrum. Figure 30 shows the frequency spectrum of the OoP deformation and the flapwise bending moment. The signal of strain gauges in the blade root from measurement and simulation no. 1 are in good agreement. The simulation shows a double peak in the range of the 2P frequency for deformation and bending moment, which can not be resolved by the measurements over the whole time





series. FFTs of shorter time series with a well-developed flapwise-vibration show that the peaks can be reproduced from those time series but over the whole time series the amplitude is not big enough to be resolved by measurements.

The FFT of IP deformation and edgewise bending moment is shown in Figure 31. The signal of strain gauges in the blade root from measurement and simulation no. 1 are in very good agreement. This proves that the simulated loads are close to the real loads in this time series. The first edgewise mode can be clearly identified out of all signals. Furthermore, the frequency

of the first flapwise mode can be seen in the IP deformation signals in simulation and measurement, as a peak in the region of the 2P frequency. A small peak at the frequency of the second flapwise mode can only be seen in the FFT of the simulated IP deformation. A reason, why this is not seen in the DIC signal, could be, that the radial measurement position is not exactly the same in simulation and measurement and that the DIC measurement point is closer to the node of the second flapwise mode.

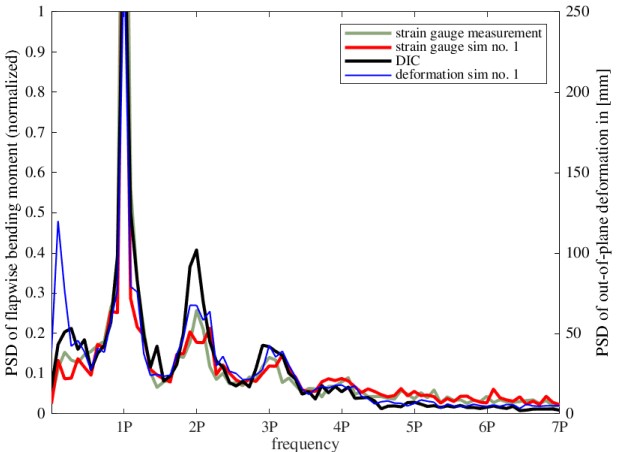

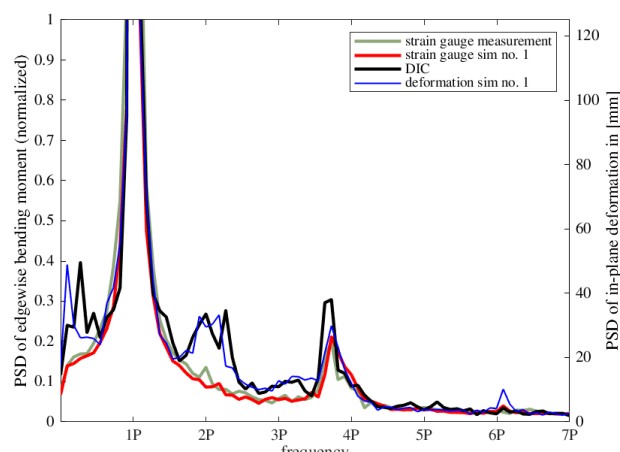

**Figure 30.** Comparison of FFT of out-of-plane strain gauge and simulation no. 1 and deformation of DIC and simulation no. 1 of Blade B

**Figure 31.** Comparison of FFT of in-plane strain gauge and simulation no. 1 and deformation of DIC and simulation no. 1 of Blade B





# 6 Conclusions

This paper summarizes shortly the functionality of DIC and the application of this innovative measurement technique to full-scale wind turbines. Furthermore, typical measurement results are shown and a comparison with measured root bending moments and simulations is evaluated.

The results show that rotor blade deformations measured with DIC qualitatively show the same trend when compared to strain gauges in the blade root for both OoP and IP. A direct comparison of measured and simulated deformations shows that

both are in very good agreement. Small deviations can be seen, especially for OoP deflections. Those deviations can occur from the statistical character of simulations. The simulations are based on the mean wind conditions of a ten minutes time series which can cause a difference between simulated loads and reality. A direct comparison of a short time series of deformation measurements with statistical simulations remains a challenge. But still the results of this paper prove that DIC is a suitable method for the validation of rotor blade deformation at full scale.

The measurement of rotor blade pitch and torsion angle with DIC in this setup clearly follows the actual pitch angle of the turbine, which validates the method on average. However, the amplitude of the dynamic torsion is higher compared to simulations. This amplitude is physically not plausible, as the expected loads of the turbine would then be different, too. The reason for this is not clear yet and will be further investigated. One approach could be to improve the experimental setup, as the present one can be considered as minimalistic in that there were only 50 speckles applied on every blade. If the size of the

speckles is reduced, the number of speckles on the blades can be increased which would come along with an improved spatial resolution of the rotor. This will be taken into account for future DIC measurements.

In summary, DIC can be considered a suitable method to measure rotor blade deformation and torsion and to validate aeroelastic simulations. The method can be easily applied on rotor blades, even if the rotor blades are already installed on the turbine. In future work, the measurement accuracy for the rotor blade torsion will be improved by optimizing the experimental

setup, in particular the speckle pattern on the blades and on the hub, as well as the measurement equipment. DIC has a great potential for the experimental validation of the simulations of rotor blade deformation and torsion of wind turbines.

*Competing interests.* The authors declare that they have no conflict of interest.

*Acknowledgements.* The autors gratefully acknowledge the financial funding from the Ministry of Science and Culture of Lower Saxony, Germany. We thank our colleagues at Siemens Gamesa and TFD for the valuable discussions concerning the results of this work. We also

thank the technical staff from Siemens Gamesa and TFD for their support during the measurement campaign.



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
