# Peer review of "Full scale deformation measurements of a wind turbine rotor in comparison with aeroelastic simulations"

_Wind Energy Science, 2020_

## Referee Comment (RC1) · Jesper Stærdahl (Referee) · 10 Mar 2020

The authors present a method using image processing for tip deformation tracking. As stated this is an important design channel for very flexible large turbine blades.

Validation of a proposed method is presented and promising results indicate that this can be used as an alternative to other deformation tracking methods.

Issus regarding torsion tracking are discussed and next steps are suggested. One of the limitations for the present method and implementation towards practical application is storage of images as it significantly limits the usability of the method. A single 10min

period as used in the present work, is too little for full design evaluation. However, principals of the method are sufficiently validated and well described and I recommend the work for publication.
* * *

---

## Referee Comment (RC2) · Anonymous Referee #2 · 12 Mar 2020

The paper explains a very promising and useful technique for measuring blade deflections. It shows experimental results obtained from a real turbine, data processing and a comparison with a numerical model.

The major issue in this paper is that the authors found the in-plane displacement to be greater than the out-of-plane one. This is in contradiction with the behavior of any turbine, and should be carefully investigated until the source of the problem is discovered.

The following remarks should also be addressed before the publication.

- It would be nice to add a few statements on how the markers on the speckle pattern are matched with the numerical model.

[Figure]

- I think that it's worth citing "Health monitoring of wind turbine blades in operation using three-dimensional digital image correlation" from Rong Wu et al.

- I understand that the illumination conditions are a challenge for DIC, but if I remember well, some authors applied phosphorescent markers, and did the test during the night.

- Almost all of the mathematical formulas are not clearly written. The authors should specify the meaning of all symbols, and the indices of the summations.

- The authors should elaborate on how the geometry of the undeformed blade is taken into account. I'm referring in particular to: cone angle, pre-bend, backward sweep and twist angle.

- Without further analyses, I would expect that most of the correlation between the out-of-place deflection, and the flapwise bending moments, is due to the 1P.

- The comparison between the strain gauges and the numerical model could be expanded by including the Damage Equivalent Loads.

- In figures 20 and 21 the authors compare the DIC measures to 9 numerical simulations, that differ by the turbulence seed. I don't think that this is a fair comparison, as the purpose of doing simulations for multiple seeds is to get accurate statistics. I would thus compute statistics for the numerical simulations (mean, standard deviation, PSD, ...) and compare them to the ones for the measures.

- Figures 22, 23, 24, 25, 28 and 29 are subject to the same problem. I think it would be better to have for each figure: - 1 curve for the mean deflection (over time) from DIC, as a function of the azimuth angle - 1 curve for the mean deflection (over time and seeds) from DIC, as a function of the azimuth angle - 2 curves for the 2 * std from the DIC - 2 curves for the 2 * std from the simulations

- I don't agree with using the FFT on noisy data, because the conditions are never constant, and the time series never periodic. I would instead use the Welch's method.

- For a future work I would recommend to run the numerical simulations using a reconstructed wind field.

Please see the attached document for additional comments.

Please also note the supplement to this comment:
https://www.wind-energ-sci-discuss.net/wes-2020-28/wes-2020-28-RC2-supplement.pdf
* * *
[Figure]

**Supplement:**

[revised manuscript text omitted]

---

## Author Comment (AC1) · 22 Apr 2020

The authors thank the reviewer for the time to read and review the paper. We appreciate your positive feedback and will work on improving the method to make it a useful tool for design evaluation.

---

## Author Response (AR1)

**1 Authors' responses**

**++ major issue ++ reviewer #2**: The major issue in this paper is that the authors found the in-plane displacement to be greater than the out-of-plane one. This is in contradiction with the behavior of any turbine, and should be carefully investigated until the source of the problem is discovered.

**Authors' response #1**: It is important to note that the results shown are for in-plane and out-of-plane vibration amplitude, and not the absolute displacement from the undeformed state. The amplitude of vibration of the out-of-plane motion can certainly be lower than the in-plane motion, as this is dependent on the turbulence level, shear, wind speed, etc. (contrary to the absolute displacement, which, as you mention, is much higher in the out-of-plane direction, in particular for wind speeds close to rated speed). In addition, the measured behavior is in accordance with the aeroelastic simulations. Therefore, the authors do not see an issue with the shown results.

**++ remark #1 ++ reviewer #2**: It would be nice to add a few statements on how the markers on the speckle pattern are matched with the numerical model.

**Authors' response #2**: From the speckle pattern, a distinctive radial position is chosen. This same radial position is chosen in the numerical model for comparison of results. This is added in the revised manuscript.

**++ remark #2 ++ reviewer #2**: I think that its worth citing "Health monitoring of wind turbine blades in operation using three-dimensional digital image correlation" from Rong Wu et al.

**Authors' response #3**: Thank you for that suggestion, we included the paper in the state of the art section.

**++ remark #3 ++ reviewer #2**: I understand that the illumination conditions are a challenge for DIC, but if I remember well, some authors applied phosphorescent markers, and did the test during the night.

**Authors' response #4**: This is a possibility and we already conducted some pre-tests with this in our laboratory. But it either reduces the time slots for measurements, as this is only feasible during the dawn without artificial illumination, or it significantly increases the effort under application of artificial illumination. In this paper, the functionality of DIC should be proven, before the effort for further measurements is increased.

**++ remark #4 ++ reviewer #2**: Almost all of the mathematical formulas are not clearly written. The authors should specify the meaning of all symbols, and the indices of the summations.

**Authors' response #5**: We completely updated this section in the revised paper and hope it is clearer now.

**++ remark #5 ++ reviewer #2**: The authors should elaborate on how

the geometry of the undeformed blade is taken into account. Im referring in particular to: cone angle, pre-bend, backward sweep and twist angle.

**Authors' response #6**: The undeformed blade geometry is not known by the university and is not necessary for the measurement of vibrational amplitudes with DIC. We measured the difference of deformation between a reference state during the measurement, which is already deformed, and the following deformed states. Thus, the measured amplitude is not the absolute displacement of the blade but a relative displacement to the reference state, thus the amplitude of vibration.

**++ remark #6 ++ reviewer #2**: Without further analyses, I would expect that most of the correlation between the out-of-place deflection, and the flapwise bending moments, is due to the 1P.

**Authors' response #7**: Yes, this is true for a large percentage of the design conditions. The turbine does not run close to any of the eigenfrequencies of the system, therefore it behaves mainly as a system undergoing a forced excitation, as you say, corresponding to 1P and its harmonics. The edgewise frequencies of the system are clearly identified.

**++ remark #7 ++ reviewer #2**: The comparison between the strain gauges and the numerical model could be expanded by including the Damage Equivalent Loads.

**Authors' response #8**: This is a good idea for a following work, but it is not the purpose of the article. This would also require to develop a transfer function between displacements and bending moments.

**++ remark #8 ++ reviewer #2**: In figures 20 and 21 the authors compare the DIC measures to 9 numerical simulations, that differ by the turbulence seed. I dont think that this is a fair comparison, as the purpose of doing simulations for multiple seeds is to get accurate statistics. I would thus compute statistics for the numerical simulations (mean, standard deviation, PSD, ...) and compare them to the ones for the measures.

**Authors' response #9**: Figures 20 and 21 were included to show some direct results of a comparison between measurement and simulations. The authors agree, that these plots should not be used for a detailed analysis which is consequently not done in the paper. Thus, in figures 22 and 23, mean and standard deviations were compared to each other. As the simulations with multiple seeds do not differ a lot in the amplitude of OoP and IP deformation, the authors think that those plots are a valid tool for a comparison.

**++ remark #9 ++ reviewer #2**: Figures 22, 23, 24, 25, 28 and 29 are subject to the same problem. I think it would be better to have for each figure: - 1 curve for the mean deflection (over time) from DIC, as a function of the azimuth angle - 1 curve for the mean deflection (over time and seeds) from DIC, as a function of the azimuth angle - 2 curves for the 2 * std from the DIC  2 curves for the 2 * std from the simulations.

**Authors' response #10**: This was done as the DIC measurement does not span a long enough period to cover "several statistic timestamps", i.e. several 10-min intervals. The authors show comparisons of those statistics in Figures 23 and 25 for single simulations and errorbars for both DIC and simulations, including +/- standard deviation. If these curves should be shown for all simulations, the plots would not be clear to read anymore.

**++ remark #10 ++ reviewer #2**: I dont agree with using the FFT on noisy data, because the conditions are never constant, and the time series never periodic. I would instead use the Welchs method.

**Authors' response #11**: We think that the FFT can also be used for noisy data, if the boundaries are moving which was the case in our plots. Nevertheless, we checked the reults for the Welchs method and agree with you, that this is the more useful method in this case. We thus replaced the FFT plots with Welch plots in the revised paper.

**++ remark #11 ++ reviewer #2**: For a future work I would recommend to run the numerical simulations using a reconstructed wind field.

**Authors' response #12**: The numerical simulations are run according to industry standard methods similar to the activities relevant for model validation during certification activities within this paper. But we agree to your comment and are currently dealing with a comparison of DIC measurements and simulations, based on a reconstructed wind field.

**2 List of changes**

**2.1 Introduction**

- added the paper of Wu et al. to the state of the art section
- changed the frequency analysis from FFT to PSD

**2.2 Experimental Setup**

- corrected stylistic errors
- explained, why the wind direction should be in the region between the two cameras
- explained why temperature, pressure and humidity are logged

**2.3 Digital Image Correlation**

- introduced the greyscale signatures $F$ and $G$ as function of the pixel position
- introduced $\sum F$ and $\sum G$ as the sum of greyscale signatures in the neighbourhood of the subset
- introduced $\chi$, the correlation function, as the difference between $F$ and $G$
- mentioned the iterative computing process to find the optimal match between $F$ and $G$
- introduced $\bar{F}_i$ and $\bar{G}_i$
- changed the color distriubtion and axis labels in Figure 8

**2.4 Determination of rotor blade deformation and torsion**

- no changes made in this section

**2.5 Results**

- mentioned the matching of the radial positions of experiment and simulations
- changed the description of the table from "figure" into "table"
- changed the description of Figure 15
- updated Figure 18 for a better comparison
- changed the FFT plots to PSD Welch plots
- updated the description of the PSD plots

**2.6   Conclusions**

- no changes made in this section

**2.7   References**

- added the paper of Wu et al.

**3  Marked-up manuscript version**

[revised manuscript text omitted]

---

## Author Response (AR2)

**1   Authors' response**

**++ overall comment ++ associate editor**: Nice paper and very interesting! However, what the paper title and abstract promise is not fully delivered in the paper. It is very interesting work but the paper seems to be more about the DIC method and measurement campaign and less about the validation. It provides an example of how comparison and validation can be done but not sufficient for a journal article on that specific topic. Generally, there should be a strong explanation for why only one time period of 5 minutes was used. Even two different time periods would bolster the analysis much more strongly.

**Authors' response #1**: Thank you for your detailed feedback. We have reformulated and clarified the abstract and introduction to introduce the novelty of this paper in comparison with the state of the art, which then justifies the presentation of an example of a comparison and validation based on one 5 minute time period. As DIC is a novel method for the detection of rotor blade deformation in the field, this technique is currently in development with potential for improvement. For example, the turbine needs to be in a stable yaw direction, which limits the length of a single time series. DIC as a new measurement method, which is currently in development and shows very promising potential for the validation of aeroelastic codes. The validation of codes based on more time series and more detailed simulations is explained in the outlook-section at the end of the paper and is subject of current investigations.

**++ detailed comment #1 ++ associate editor**: Abstract is too long. Consider shortening and focus on key contributions. The abstract is not a place to provide background and introduction nor future work. The second paragraph has some repetitive content to the first.

**Authors' response #2**: We agree with you and shortened the abstract according to your suggestion.

**++ detailed comment #2 ++ associate editor**: Opening statement not substantiated by a reference. Slightly odd in wording. Why does reduce LCOE lead to larger rotors? This is not self-evident. A reference or elaboration would help. Similar on second sentence too  relative mass to what? Refer to square cube law perhaps  veers et al 2019 as well as the longer IEA Wind report dykes et al 2019 go into detail on this: https://www.nrel.gov/docs/fy19osti/72437.pdf. Sorry to be picky but as an opening to the paper, it should be stronger.

**Authors' response #3**: This is a very valuable remark. We have reformulated the introduction and added more references to this.

**++ detailed comment #3 ++ associate editor**: The uniqueness of the present work compared to Winstroth et al 2014 would be nice to call out explicitly directly after that work is cited. See overall comments, the novelty of this paper needs to be further espoused and built up if the use of a single 5 minute period of the measurement campaign (in results section) is to be justified.

**Authors' response #4**: This is a very important remark, thank you for

pointing this out. We changed the structure and formulation of the introduction part and further elaborated on the differences and novelties in comparison with the work that is already published.

**++ detailed comment #4 ++ associate editor**: Since the article at the beginning uses validation of aeroelastic codes as motivation for the work, it is underwhelming to simply report the solver being used without more details on the simulation setup. Though BHawC is a property code, minimum details about its phsycial modeling characteristics should be set up  what type of aerodynamic solver, structural solver, etc? I believe you are using the statistics of the wind conditions of the site to produce the wind field that is fed to the solver but it is not very clear since there really isnt a good description overall of the experimental setup.

**Authors' response #5**: This is true, so we added more information on the solver and the simulations. Statistics of wind conditions were used for the simulations, which we emphasized more clearly.

**++ detailed comment #5 ++ associate editor**: I agree with the reviewer 2 comments on the turbulent seeds. It is very odd to show simulations from multiple turbulent seeds versus statistics. Even if you are simply trying to show the comparison, I would still show the simulation results with a mean and standard deviation or percentile band  as that is common practice. The same recommendation goes for figures 21-24, the graphs are hard to read as it is and you could mitigate this simply by providing the mean across the simulations with a standard deviation band around it

**Authors' response #6**: We changed the plots according to your suggestion and completely agree with you, that the plots are much better to read now.

**++ detailed comment #6 ++ associate editor**: Psd is very interesting but it would be nice to see the analysis completed for more than one sample.

**Authors' response #7**: The PSD was updated to involve not only one simulation but the mean of all simulations.

**2 List of changes**

**2.1 Abstract**

- shortened and clarified the abstract

**2.2 Introduction**

- elaborated on the context between reduced costs and the size of the rotor
- added more references to underline the statements
- further elaborated on the novelty of this paper in comparison with already published work
- introduced the use of a single five minutes measurement time series as an exemplary comparison between measurements and simulations

**2.3 Experimental Setup**

- no changes made in this section

**2.4 Digital Image Correlation**

- no changes made in this section

**2.5 Determination of rotor blade deformation and torsion**

- no changes made in this section

**2.6 Results**

- updated the description of aeroelastic code and the use of statistical wind conditions
- joined the single simulations of multiple turbulent seeds to one resulting curve with mean and standard deviation for all plots including simulation results

**2.7 Conclusions**

- small changes in wording
- further elaborated on future work for the validation of aeroelastic codes based on more series of measurements

**2.8 References**

- added more references for the introduction and for aeroelastic simulations

**3   Revised manusript**

[revised manuscript text omitted]